

# Physicochemical properties-based hybrid machine learning technique for the prediction of SARS-CoV-2 T-cell epitopes as vaccine targets

Syed Nisar Hussain Bukhari[1], E. Elshiekh[2] and Mohamed Abbas[3]

[1] National Institute of Electronics and Information Technology (NIELIT), Srinagar, Jammu and Kashmir, India
[2] Department of Radiological Sciences, College of Applied Medical Sciences, King Khalid University, Abha, Saudi Arabia
[3] Electrical Engineering Department, College of Engineering, King Khalid University, Abha, Saudi Arabia

## ABSTRACT

Majority of the existing SARS-CoV-2 vaccines work by presenting the whole pathogen in the attenuated form to immune system to invoke an immune response. On the other hand, the concept of a peptide based vaccine (PBV) is based on the identification and chemical synthesis of only immunodominant peptides known as T-cell epitopes (TCEs) to induce a specific immune response against a particular pathogen. However PBVs have received less attention despite holding huge untapped potential for boosting vaccine safety and immunogenicity. To identify these TCEs for designing PBV, wet-lab experiments are difficult, expensive, and time-consuming. Machine learning (ML) techniques can accurately predict TCEs, saving time and cost for speedy vaccine development. This work proposes novel hybrid ML techniques based on the physicochemical properties of peptides to predict SARS-CoV-2 TCEs. The proposed hybrid ML technique was evaluated using various ML model evaluation metrics and demonstrated promising results. The hybrid technique of decision tree classifier with chi-squared feature weighting technique and forward search optimal feature searching algorithm has been identified as the best model with an accuracy of 98.19%. Furthermore, K-fold cross-validation (KFCV) was performed to ensure that the model is reliable and the results indicate that the hybrid random forest model performs consistently well in terms of accuracy with respect to other hybrid approaches. The predicted TCEs are highly likely to serve as promising vaccine targets, subject to evaluations both *in-vivo* and *in-vitro*. This development could potentially save countless lives globally, prevent future epidemic-scale outbreaks, and reduce the risk of mutation escape.

Corresponding author
Syed Nisar Hussain Bukhari,
nisar@nielit.gov.in

# INTRODUCTION

The emergence of the novel beta coronavirus, namely Severe Acute Respiratory Syndrome Coronavirus 2 (SARS-CoV-2), took place in China's Hubei province in late 2019. This virus

is accountable for the onset of coronavirus disease 2019 (COVID-19) and was officially designated as a pandemic by the World Health Organization (WHO) on March 11, 2020, according to *Chakraborty et al. (2020)*. The global impact of the COVID-19 pandemic has been profound, affecting countries socially and economically. It is anticipated that the virus will continue to be a significant aspect of our daily lives for an extended period. Worldwide, there was a 4% increase in the number of new cases between December 11, 2023, and January 7, 2024, compared to the preceding 28-day period, totaling more than 1.1 million new cases. JN.1 currently stands as the predominant variant of interest (VOI), detected in 71 countries, constituting approximately 66% of sequences in week 52, a notable rise from around 25% in week 48. Conversely, there was a 26% reduction in new deaths during the same 28-day period, with 8,700 reported fatalities. As of February 24th, 2024 WHO has reported 703,601,632 confirmed cases of COVID-19 worldwide, with 6,985,676 deaths (*Worldometer, 2024*). Mutations in the spike protein of SARS-CoV-2 impact both the ability to infect and the efficacy of vaccines (*Ishige, 2024*). Similar to many RNA viruses, coronaviruses undergo rapid evolution, which unfolds over months or years and is frequently noticeable and quantifiable. This evolutionary process unfolds concurrently with virus transmission events and ecological changes, including alterations in the population of infectious individuals, immunity patterns, and human movement (*Markov et al., 2023*). Consequently, evolutionary, ecological, and epidemiological phenomena influence one another, a characteristic shared by RNA viruses.

Regarding taxonomy, *Coronaviruses* fall under the Orthocoronavirinae subfamily within the Coronaviridae family, a constituent of the *Cornidovirineae* sub-order within the *Nidovirales* order, as outlined by *Pal et al. (2020)*. SARS-CoV-2, akin to other coronaviruses, exhibits a morphology characterized by a single-stranded, positive-sense RNA molecule, comprising around 29,900 nucleotides (per NCBI Reference Sequence: NC_045512.2), as indicated by *Su et al. (2016)*. The estimated genome size of SARS-CoV-2 ranges between 26–32 kb. The genetic material of SARS-CoV-2 comprises 14 ORF (open reading frame) sequences, generating a total of 29 proteins. Among these proteins, four are structural, namely S (spike), E (envelope), M (membrane), and N (nucleocapsid), while the remaining 16 are non-structural proteins (nsp). Additionally, there are nine accessory proteins, including the RNA-dependent RNA polymerase (RdRp), also known as nsp12, as highlighted by *Khailany, Safdar & Ozaslan (2020)* and *Liu et al. (2014)*. Vaccination has emerged as the primary strategy for preventing SARS-CoV-2 infection. Currently, many of the available vaccines for SARS-CoV-2 are whole-organism vaccines, including live attenuated and inactivated vaccines. However, these types of vaccines can be costly to manufacture, necessitate cultivation of the infectious agent, and may induce vaccine-associated illnesses in hosts (*Roper & Rehm, 2009*). Moreover, they may not be suitable for individuals with compromised immune systems and their storage requires careful temperature control (*Cai et al., 2021*). Therefore, there has been a shift towards the development of Peptide-Based Vaccines (PBVs), which involve the identification and chemical synthesis of immunodominant peptides known as T-cell epitopes (TCEs) which are capable of inducing specific immune responses against the pathogen (*Rosendahl Huber et al., 2014*). Selective extraction of antigenic components involves isolating only

those segments of a protein capable of triggering an immune response, discarding non-essential elements (*Seder, Darrah & Roederer, 2008*). PBVs offer several advantages over conventional vaccines, including fewer side effects, ease of manufacturing, absence of whole pathogen components, improved specificity, stability, sustainability, and shorter production times (*Grifoni et al., 2020*). Despite these advantages, PBVs have received less attention, and their potential for improving vaccine safety and immunogenicity remains largely untapped (*Lee et al., 2021*).

The identification of immunodominant epitopes for PBV design through wet lab experiments is a difficult, expensive, and time-consuming process. Nevertheless, machine learning (ML) methods can predict these epitopes with considerable precision, thereby expediting vaccine development and reducing costs when contrasted with wet-lab techniques (*Bukhari et al., 2021*). It is essential to highlight the pivotal role of T-cells in adaptive immunity, contributing crucial helper functions to various arms of the immune system and playing a vital role in controlling, clearing, and providing protection against a wide range of viral infections, as emphasized by *Moss (2022)*. Evidence drawn from previous studies on SARS-CoV-1 and MERS indicates that T-cells may serve as primary mediators for disease control (*Liu et al., 2019*). Research suggests that the T-cell response to SARS-CoV-2 infection targets multiple antigens of the virus, with prominent reactivity against spike and nucleocapsid proteins (*Zhang et al., 2020*), indicating that T-cells are well-equipped to handle emerging viral variants (*Niessl, Sekine & Buggert, 2021*).

In the pursuit a potential vaccine against SARS-CoV-2, this study introduces a novel approach to predict the TCEs of the virus. The method employs hybrid machine learning techniques, incorporating the physicochemical properties of peptides. The epitopes predicted through this approach could potentially serve as vaccine targets for designing the PBVs. The proposed model is poised to assist the scientific community in predicting novel and immunodominant TCEs of SARS-CoV-2.

## Background

The SARS-CoV-2 virus underwent frequent genetic changes and mutations which caused widespread concerns all over the world (*O'Toole & Hill, 2024*). To guard against the emergence of these variations, there are two potential approaches: adjusting the formulation of current vaccines or creating an entirely new vaccine (*Shang et al., 2020*; *Roper & Rehm, 2009*). Given the urgency of the situation, PBVs stand out as a promising alternative. This option is advantageous due to its cost-effectiveness, shorter production timeline, safety, and potential to enhance both immunogenicity and cross-reactivity (*Moss, 2022*). Additionally, identifying SARS-CoV-2 epitopes using existing methods such as the NetMHC is often probabilistic, with limitations in predicting whether a peptide is an epitope or not (*Nielsen et al., 2003*). CTLpred can make deterministic predictions, but is restricted to peptides of length up to 9-mers only (*Bhasin & Raghava, 2004*). To address these limitations, this study introduces a hybrid ML model that can predict TCEs directly and of varying lengths (>9 amino acids).

## Contributions

This research introduces several innovative contributions. Primarily, the study involves the development and evaluation of 27 hybrid ML techniques. These techniques are constructed through permutations and combinations of three classification algorithms, three feature weighting algorithms, and three feature selection techniques, all geared towards predicting TCEs of the SARS-CoV-2 virus. Second, we introduced a new feature extraction technique that extracts the physicochemical properties of peptides at the amino acid level. Third, we used heuristic and greedy search techniques to identify optimal features for training the models after extracting features from peptide sequences. Fourth, the focus of this study was achieving high accuracy in predicting TCEs and the proposed hybrid techniques showed promising results in terms of accuracy. The proposed models underwent evaluation based on various parameters such as area under the curve (AUC), sensitivity, specificity, Gini, F-score, and MCC. The results indicate that the combination of decision tree (DT) with chi-squared and forward search emerges as the most accurate and reliable predictive method for forecasting TCEs of SARS-CoV-2. Additionally, K-fold cross-validation (KFCV) was conducted, affirming the consistency and reliability of the proposed model for TCE prediction across all folds. The predicted epitopes have the potential to act as primary targets for designing a PBV.

The subsequent sections of the manuscript are organized as follows: The related work section presents a comprehensive literature review. The materials and methods section details the proposed methodology. The results section showcases the obtained results. Finally, the conclusion section concludes the manuscript.

## RELATED WORK

Numerous research efforts have been dedicated to identifying TCEs of the SARS-CoV-2 virus for the development of PBVs. *Awad et al. (2022)* employed immunoinformatics tools to scrutinize entire viral protein sequences, pinpointing SARS-CoV-2 epitopes interacting with prevalent human leukocyte antigen (HLA) alleles among the Egyptian population. The analysis predicted seven potential vaccine subunits based on available SARS-CoV-2 spike and ORF1ab protein sequences, unveiling two novel epitopes, RDLPQGFSA and FCLEASFNY, with high docking scores and antigenicity responses with both MHC-I and TCR. *Foix et al. (2022)* utilized the NetMHCIIpan EL algorithm to forecast HLA class I epitopes within the SARS-CoV-2 proteome across 2,915 human alleles from 71 families. Their results displayed substantial differences in epitopic profiles among allele families, emphasizing genetic variability in protective capacity. The study identified up to 1,222 epitopes associated with twelve supertypes, covering 90% of the population (*Bukhari, Webber & Mehbodniya, 2022*; *Vita et al., 2019*) developed an ensemble machine learning model for predicting TCEs of SARS-CoV-2, achieving an accuracy of 98.20% and an AUC of 0.991. Their model outperformed existing TCE prediction techniques, showcasing superior performance. *Mahajan et al. (2021)* utilized a novel TCR-binding algorithm to identify CD8 TCEs in the spike antigen, inducing strong T-cell activation even in donors unexposed to SARS-CoV-2, suggesting pre-existing CD4 and CD8 T-cell immunity. *Rencilin et al. (2021)* identified 2,604 epitopes as potent binders to MHC class I molecules,

narrowing down to 50 peptides with potential CTL epitopes. *Meyers et al. (2021)* utilized immunoinformatics to identify TCEs from the envelope, membrane, and spike portions of SARS-CoV-2, revealing strong T-cell responses for 97% of tested epitopes. *Fatoba et al. (2021)* predicted CD8, CD4, and linear B-cell epitopes, identifying 19 CD8 and 18 CD4 epitopes for potential use in PBVs. *Naz et al. (2020)* and *Grifoni et al. (2020)* screened SARS-CoV epitopes, sharing genetic similarity with SARS-CoV-2, suggesting their utility in vaccine development. *Baruah & Bose (2020)* identified B-cell and CTL epitopes from the surface glycoprotein of SARS-CoV-2, some considered as potential vaccine targets. *Crooke et al. (2020)* utilized computational methods, yielding 41 B-cells and six TCEs for potential inclusion in a PBV. *Dong et al. (2020)* developed an immunoinformatics-based multi-epitope vaccine for COVID-19 treatment and prevention, fusing B-cell, CTL, and HTL epitopes of SARS-CoV-2 proteins. The study conducted by *Bravi (2024)* elaborates on the utilization of ML in rational vaccine design, particularly in identifying B and T cell epitopes and protection correlates. Various ML models have been explored and their data dependencies, emphasizing the potential of interpretable ML in enhancing immunogen identification and scientific comprehension. The study by *Yang, Bogdan & Nazarian (2021)* proposes DeepVacPred, an *in silico* deep learning approach for designing a multi-epitope vaccine against SARS-CoV-2. Using immunoinformatics and deep neural network strategies, 26 potential vaccine subunits are predicted from the viral spike protein sequence. After rigorous evaluation, 11 optimal subunits are selected for vaccine construction. The resulting 694aa multi-epitope vaccine exhibits promising features and is capable of targeting recent RNA mutations of the virus. The study by *Manavalan, Basith & Lee (2022)* reviews the landscape of ML algorithms for predicting antiviral peptides (AVPs) against SARS-CoV-2, emphasizing the need for more accurate and efficient prediction tools in the context of the COVID-19 pandemic. By systematically evaluating existing ML predictors and comparing their performances, the study offers insights into the strengths and limitations of current approaches. The findings aim to guide the development of next-generation computational tools for identifying therapeutic peptides against COVID-19, thus contributing to the ongoing efforts in combating the disease. Another study conducted by *Ghosh, Larrondo-Petrie & Pavlovic (2023)* explores role of artificial intelligence (AI)/ML techniques in overcoming vaccine development challenges and highlights recent advancements in COVID-19 drug and vaccine development through AI-driven approaches.

# MATERIALS AND METHODS

In this section, our methodology for predicting TCEs of the SARS-CoV-2 virus, essential for the design of a PBV will be explicated. The methodology encompasses the application of three distinct feature weighting (FW) techniques, three optimal feature selection (OFS) techniques and three classification algorithms. Through the systematic exploration of permutations and combinations of these methodological components, we have introduced hybrid approaches tailored for the accurate prediction of TCEs associated with the SARS-CoV-2 virus. The methodology adopted in the current study is explained through the following sub-sections.

## Retrieval of peptide sequences

To obtain the peptide sequences for this study, we accessed the Immune Epitope Database (IEDB) repository (*Vita et al., 2019*). Specifically, we obtained two comma-separated value (CSV) files, wherein one of the files consisted of TCE sequences while the other file contained non-T-cell epitopes (NTCEs). All sequences are experimentally verified sequences and are of linear type. To perform binary classification, a target variable named "Class" was added to both CSV files, with epitope sequences.

## Feature extraction

Each peptide sequence is characterized by its constituent amino acids and possesses physicochemical properties, which act as features in this study. To extract these features we performed feature extraction (FE) using the peptides (*Osorio, Rondón-Villarreal & Torres, 2015*) and peptider (*Hofmann, Hare & GGobi Foundation, 2015*) (Evaluation of Diversity in Nucleotide Libraries (R Package Peptider Version 0.2.2), 2015) packages of R programming language (*R Core Team, 2013*). Prior to applying FE, the duplicate peptide sequences were removed. The FE technique was employed separately to each CSV file (one containing TCE sequences and the other containing NTCE sequences) and a high-dimensional dataset consisting of 108 features for each sequence was produced. The two CSV files were merged into one CSV file. The merging of these CSV files facilitated the creation of a unified dataset ensuring that both TCEs and NTCEs labelled data instances are present in one single file. This comprehensive dataset enabled the training and evaluation of ML models on sequences belonging to both the classes *i.e.*, T-cell epitope (TCE) sequences and non-T-cell epitope (NTCE) sequences. Table 1 enumerates the physicochemical properties employed in this investigation, and Table 2 provides a summary of the dataset post feature extraction (FE) procedures.

## Feature selection

Feature selection (FS) constitutes a crucial phase in the machine learning (ML) process, wherein the objective is to choose the most pertinent features from a dataset for constructing a predictive model. The overarching aim of FS is to reduce the number of features in the dataset while preserving those that are most instrumental in enhancing the predictive accuracy of the model (*Gupta & Rana, 2019*). This process not only helps in reducing the computational cost and overfitting but also improves the generalization of the model. There are certain criteria that should be considered when selecting features. First, the selected features should be relevant to the problem being solved and should have a significant impact on the model's performance. Second, the features should be independent of each other to avoid redundancy and multicollinearity, which can affect the stability and interpretability of the model. Ultimately, the chosen features should exhibit robustness against noise and outliers present in the data. Given the dataset's high dimensionality, comprising 108 features, it becomes imperative to assign weights to all features and then identify the optimal subset. This process aims to augment the efficiency of the machine learning (ML) model. The subsequent subsections offer a detailed overview

**Table 1 Physicochemical properties used.**

| Physicochemical property | Count | Notation |
|---|---|---|
| Aliphatic index | 1 | F1 |
| Boman index | 1 | F2 |
| Insta index | 1 | F3 |
| Probability of detection | 1 | F4 |
| Hmoment index | 2 | F5_1, F5_2 |
| Molecular weight | 2 | F6_1, F6_2 |
| Peptide charge for 45 scales | 45 | F7_1 to F7_45 |
| Hydrophobicity at 44 scales | 44 | F8_1 to F8_44 |
| Isoelectric point for 9 pKscale | 9 | F9_1 to F9_9 |
| Kidera factors | 1 | F10 |
| aaComp | 1 | F11 |

**Table 2 Overview of the dataset.**

| Peptide sequence | F1 | F2 | ——— | F10 | F11 | Class |
|---|---|---|---|---|---|---|
| RLANECAQV | 38.12 | 0.4592 | ——— | −1.287 | 2.721 | 1 |
| EILDITPCSFG | 46.64 | 0.8736 | ——— | −0.102 | −0.176 | 1 |
| RVVRSIFSR | 162.10 | 1.238 | ——— | −0.289 | −0.091 | 0 |
| TLADAGFIK | 109 | −4.102 | ——— | −0.178 | −2.371 | 0 |

of the procedures involved in assigning feature weights and determining the optimal feature subset.

## Feature weighting

Feature weighting is a technique in ML that involves assigning weights to the features to adjust their influence on the model's output. The goal of FW is to give more importance to the relevant features and less importance to the irrelevant ones, thereby improving the accuracy and robustness of the model (*Niño-Adan et al., 2021*). In this study, three distinct FW techniques namely Information Gain (IG), Gain Ratio (GR) and Chi-Squared (ChS) from the FSelector package in R were employed to assign weights to the features (*Romanski, Kotthoff & Schratz, 2023*). A brief description of each technique is given as follows.

1) *Information Gain:* IG measures how much information is gained when a feature is used to split the data. Higher values indicate that a feature is more informative in distinguishing the classes. The function prototype of IG is given as information.gain(x, y, …). The function takes two mandatory parameters x and y, representing the feature and class variables respectively. The optional tuning parameters represented by three dots are N, the number of bins to discretize numeric variables, and FUN, the function to be used for discretization.

2) **Gain Ratio:** GR is an enhancement of IG that accounts for the number of possible splits a feature can have. It normalizes the IG by the intrinsic information of the feature, resulting in a more balanced measure. The function gain.ratio(x, y, …) takes the same parameters as IG

3) **Chi-Squared:** ChS tests the independence between a feature and the class variable. It measures the difference between the observed and expected frequencies of each class in each category of the feature. Higher values indicate that a feature is more related to the class variable. The function chi.squared(x, y …) takes the same parameters as IG.

IG was chosen for its ability to measure the information gained by a feature when used to split the data, making it particularly suitable for identifying informative features. GR, an enhancement of IG was selected to normalize the IG by the intrinsic information of the feature, providing a more balanced measure and reducing bias towards features with a large number of possible splits. ChS was deemed appropriate for testing the independence between a feature and the class variable, which is essential for identifying features that are significantly related to the target variable. It is important to note that all the function prototypes for these techniques take two mandatory parameters, x and y, representing the feature and class variables respectively, and some optional tuning parameters. IG and GR measure the amount of information gained by a feature, with GR normalizing the IG. ChS tests the independence between a feature and the class variable.

## Finding the optimal subset of features

After assigning weights to the features, it is crucial to determine the optimal feature subset (OFS) for building an accurate and efficient ML model. The quest for the optimal feature subset entails the selection of a subset from a larger pool of available features. This subset is characterized by its ability to deliver the most effective predictive performance for the ML model (*Kang & Kim, 2016*). This procedure holds particular significance in high-dimensional datasets, where the abundance of features can be substantial and may detrimentally affect the model's performance if not effectively managed. To obtain the optimal feature subset, three different effective techniques are employed in this study, namely hill-climbing search, forward search, and backward search (*Więckowski, Kizielewicz & Kołodziejczyk, 2020*). A brief explanation of each optimal feature searching techniques employed in this study is given next.

1) **Hill climbing search (HcS):** HcS is a heuristic optimization technique that iteratively evaluates the performance of different feature subsets and selects the subset that provides the highest accuracy. Starting with an initial feature subset, HcS evaluates the performance of all possible one-feature additions and selects the feature subset that provides the best improvement in accuracy. The process is repeated until no further improvements are possible.

2) **Forward search (FwS):** FwS represents a wrapper-based feature selection technique commencing with an empty feature subset. It systematically adds features in iterations, incorporating those that contribute the most significant accuracy improvement. FwS

assesses the performance of all conceivable feature additions and ultimately chooses the feature subset yielding the most substantial enhancement in accuracy. This iterative process continues until no further improvements are attainable.

3) **Backward search (BwS):** BwS is similar to FwS except that it starts with the full feature subset and iteratively removes features that provide the least improvement in accuracy. BwS evaluates the performance of all possible feature deletions and selects the feature subset that provides the best improvement in accuracy. The process is repeated until no further improvements are possible

The outputs obtained from the three aforementioned optimal feature selection techniques are then inputted into three diverse ML algorithms, namely decision trees, neural networks, and random forests, which are further elaborated in the subsequent sub-section.

## Machine learning classifiers used

The final hybrid techniques were formulated by incorporating three widely employed classification methods: decision tree (DT), random forest (RF) and neural network (NN). These techniques were integrated into the final classification step to facilitate a comparative study. Each method employs its intrinsic functionality for data classification, and the inclusion of these diverse classification techniques serves the purpose of utilizing three distinct approaches in the final phase of the proposed techniques. Typically, the DT algorithm employs a greedy approach for its classification task (*Quinlan, 1986*). DT partition the feature space into a set of disjoint regions, with each region corresponding to a unique class label. It then recursively splits the feature space based on feature values, aiming to maximize information gain or minimize impurity at each split point. On the other hand, the RF technique is an ensemble of decision trees, where multiple trees are generated, and the final classification is based on majority voting (*Liaw & Wiener, 2002*). Each tree in the forest is trained on a bootstrapped sample of the training data, and at each split point, a random subset of features is considered. Random forests mitigate overfitting and improve generalization performance by aggregating the predictions of multiple trees. NNs are a class of models inspired by the biological structure of the brain's neural networks. They consist of interconnected layers of nodes (neurons) that process input data through a series of weighted connections and activation functions. The classification output of NN depends on the activation function used to build the network (*Riedmiller & Braun, 1993*). Table 3 provides an overview of the ML classifiers used along with their corresponding tuning parameters. The brief description of each classifier is provided in Supplemental Note S1.

## Methodology

This section presents a summary of the methodology employed to create the proposed hybrid ML techniques. These models have been developed by combining three FW techniques, three OFS techniques, and three classification methods through permutation and combination approaches. Specifically, we employed three distinct FW techniques,

**Table 3  ML classifiers used.**

| Classifier | Method | Package | Tuning parameter |
|---|---|---|---|
| DT | rpart | rpart | (usesurrogate = 0, maxsurrogate = 0) |
| RF | Random Forest | Random Forest | ntree = 1,500, mtry = 10 |
| NN | nnet | nnet | Size = 10 |

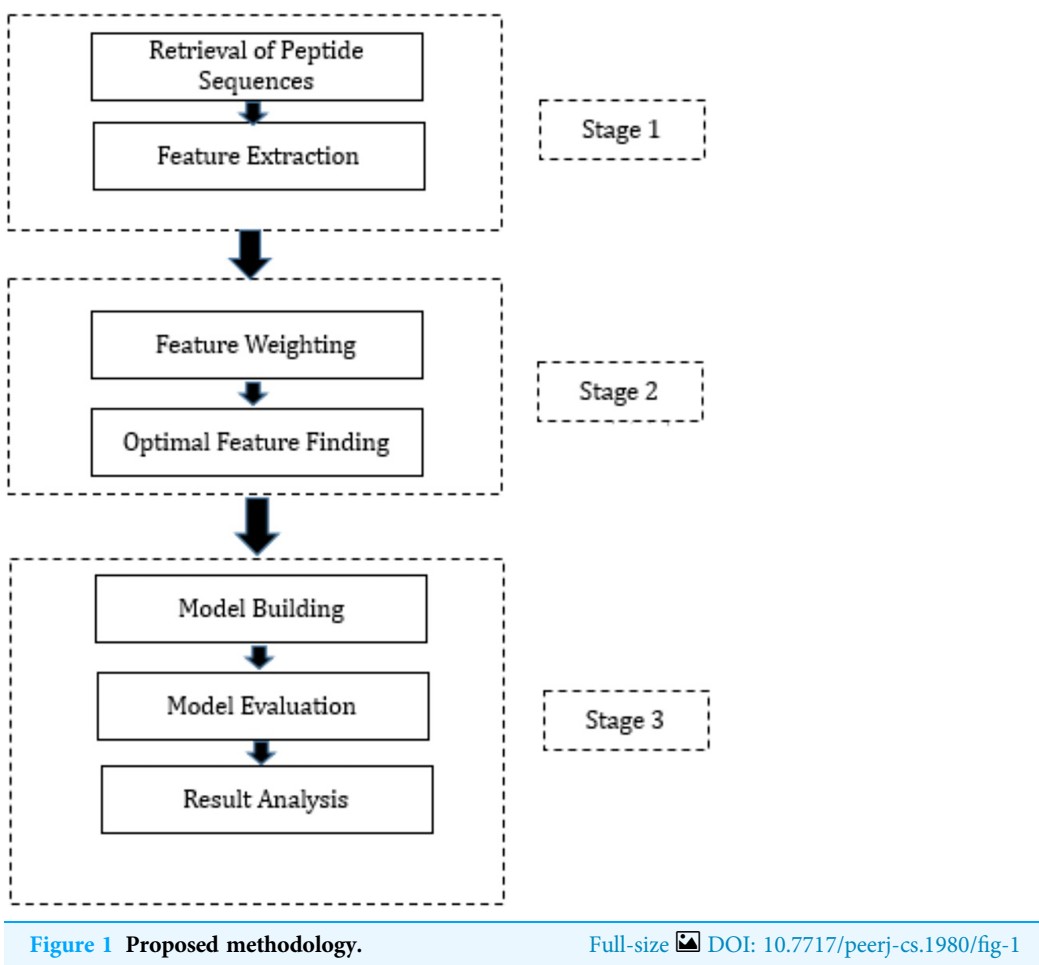

**Figure 1  Proposed methodology.**

namely Information Gain, Gain Ratio, and Chi-Squared, to assign weights to the features. These FW techniques were selected based on their ability to evaluate the relevance of features in distinguishing between different classes. Following the FW process, we employed three OFS techniques: hill-climbing search, forward search, and backward search, to determine the optimal feature subset. Each OFS technique iteratively evaluates different feature subsets to identify the subset that provides the best improvement in accuracy. Subsequently, the output from the FW and OFS techniques was integrated into three diverse ML algorithms: DT, RF, and NN. These ML algorithms were chosen for their ability to handle classification tasks and their distinct approaches to data classification. The

workflow is presented in Fig. 1 which outlines the process. The proposed hybrid techniques consist of three stages and are outlined below.

Stage 1: During this phase, the peptide sequences of SARS-CoV-2 were acquired from IEDB in CSV format. Following this, feature extraction was executed on these sequences, utilizing the feature extraction techniques provided in the "peptides" and "peptider" packages of the R programming language. This process led to the generation of high-dimensional data, with 108 features extracted for each peptide sequence. Subsequently, the data produced during the feature extraction process from various CSV files were amalgamated into a unified CSV file. It is noteworthy that this phase constitutes a standardized procedure adopted by all hybrid techniques proposed in this study.

Stage 2: In Stage 2, several crucial steps were taken, encompassing the assignment of weights to extracted features and the determination of the optimal feature subset. Post FE, the subsequent phase involves assigning weights to each feature to gauge its relative importance. To accomplish this, three distinct FW techniques were employed. Following the FW process, the subsequent step encompassed determining the optimal feature subset, achieved through three different techniques. The output of each FW technique was separately fed into each OFS technique, and the resulting output from each combination was employed in Stage 3 of the research. Precisely, these outputs were utilized to train three distinct classification algorithms. The optimal features identified by various FS methods are outlined in Table S1. Equations (1) to (3) elucidate the model formulas, with the dependent variable "Class" and its corresponding independent variables for model training via Hill climbing, forward search, and backward search techniques, respectively.

$$\text{Class} \sim f\ (F8\_13, F8\_27, \ldots\ldots\ldots\ldots, F7\_25, F11\_7) \tag{1}$$

$$\text{Class} \sim f\ F8\_47, F8\_4, \ldots\ldots\ldots\ldots., F7\_29, F7\_28) \tag{2}$$

$$\text{Class} \sim f\ (F9\_5,\ F1, \ldots\ldots\ldots\ldots, F7\_31, F8\_22) \tag{3}$$

Equation 1 represents the model formula for training a predictive model using the HcS technique for feature selection. The dependent variable 'Class' is predicted based on a set of independent variables (features) denoted by F8_13, F8_27, …, F7_25, F11_7. Equation 2 represents the model formula for training a predictive model using the FwS technique for feature selection. Similar to Eq. (1), it predicts the 'Class' based on a specific subset of independent variables. Equation 3 represents the model formula for training a predictive model using the BwS technique for feature selection. Again, it predicts the 'Class' based on a subset of independent variables.

Stage 3: The initiation of Stage 3 involves partitioning the dataset, which contains the optimal features derived from each combination of FW and OFS techniques, into training and testing sets. The split ratio was configured at 80:20, allocating 80% of the dataset for model training and reserving the remaining 20% for predictions. Subsequent to training the models with the designated training datasets, their performance was validated using evaluation parameters detailed in the model evaluation section next. To ensure the reliability of the trained models, K-fold cross-validation was executed.

## MODEL EVALUATION

Model evaluation is an essential component of any ML workflow to evaluate the performance of the ML model so that it can accurately generalize to new and unseen data (*Alpaydin, 2010*). In this study, a variety of assessment metrics have been employed, including accuracy, sensitivity, specificity, precision, AUROC, F1-score, and Mathews correlation coefficient (MCC) (*Bukhari, Jain & Haq, 2022*) and (*Zhu, 2020*). The consistency and robustness of the proposed techniques have additionally been evaluated through K-fold cross-validation (KFCV). It is important to note that ML model evaluation is an iterative process (*Khanna & Rana, 2017*). The results of the evaluation can inform adjustments to the model, such as changes to hyperparameters, feature selection, or data pre-processing. This process continues until the desired level of performance is achieved. Equations 4 to 9 define the metrics used in this study for model evaluation where TP: True Positive, TN: True Negative, FP: False Positive and FN: False Negative (*Powers, 2020*).

$$\text{Accuracy} = (TP + TN) / (TP + TN + FP + FN) \tag{4}$$

$$\text{Sensitivity} = TP / (TP + FN) \tag{5}$$

$$\text{Specificity} = TN / (TN + FP) \tag{6}$$

$$\text{Precision} = TP / (TP + FP) \tag{7}$$

$$\text{F1 score} = 2 * (\text{precision} * \text{recall}) / (\text{precision} + \text{recall}) \tag{8}$$

$$\text{MCC} = \frac{(TP * TN - FP * FN)}{\text{Sqrt}((TP + FP) * (TP + FN) * (TN + FP) * (TN + FN))} \tag{9}$$

The above Eqs. (4)–(9) define the commonly used evaluation metrics for assessing the performance of predictive models *i.e.*, accuracy, sensitivity, specificity, precision, F1-score and MCC.

### Area under the ROC curve

The ROC curve illustrates the relationship between the true positive rate (TPR) and the false positive rate (FPR) at different threshold settings (*Bradley, 1997*). The area under the ROC curve (AUROC) condenses this curve into a single value, serving as a comprehensive metric that summarizes the model's efficacy in distinguishing between positive and negative classes.

### K fold cross-validation

KFCV, or K-fold cross-validation, is a technique employed to assess the consistency and robustness of a model by dividing the original dataset into K subsets or folds of approximately equal size (*Refaeilzadeh, Tang & Liu, 2009*). The model undergoes training and evaluation K times, utilizing each fold once for evaluation while employing K-1 folds for training, as illustrated in Fig. 2. The performance metrics obtained from the K evaluations are then averaged, providing an estimate of the model's generalization performance.

The selection of evaluation metrics in this study was based on their relevance to assessing the performance of predictive models for T-cell epitopes (TCEs) of the SARS-

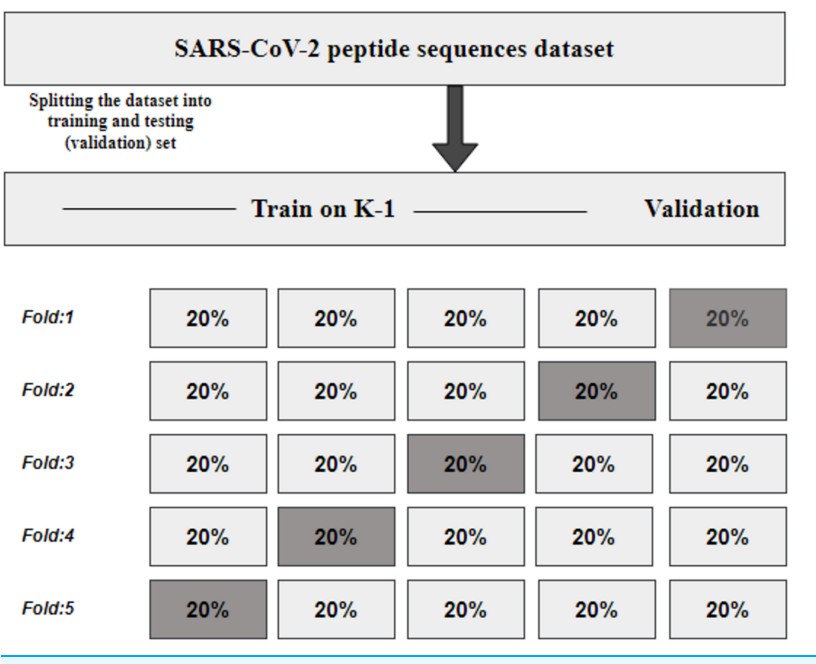

**Figure 2  KFCV technique.**

CoV-2 virus. In particular, the AUROC and MCC were chosen due to their effectiveness in evaluating binary classification models, particularly in the context of imbalanced datasets and the need for robust performance measures in biomedical applications. AUROC is a comprehensive metric that summarizes the model's ability to discriminate between positive and negative classes across different threshold settings. Given the importance of accurately identifying TCEs, AUROC provides valuable insights into the overall discriminatory power of the predictive models. MCC, on the other hand, is a balanced measure that takes into account both true positives, true negatives, false positives, and false negatives, making it suitable for assessing the overall performance of binary classification models, especially in scenarios with imbalanced class distributions.

## RESULTS

This section elucidates the outcomes derived from the implementation of diverse hybrid techniques on a high-dimensional dataset comprising 108 features extracted from the peptide sequence of SARS-CoV-2. To ascertain the most effective hybrid technique, a comprehensive comparative analysis was conducted among the various hybrid methodologies employed in this investigation. This comparison was conducted with due consideration to the aforementioned evaluation parameters. Table 4 displays the accuracies of the various hybrid approaches used in this study.

It is evident from the Table 4 that random forest yields exceptional results with all FW and OFS techniques, delivering an accuracy of more than 93% in all scenarios. Notably, the hybrid approach of gain-ratio and hill climbing produces the highest accuracy of 97.29% for RF models. In terms of accuracy, DT is the second-best model with all FW and optimal feature-finding techniques, achieving an accuracy as low as 72.56%. However, it attains the

**Table 4 Accuracies achieved by different hybrid models.**

| FW Algorithm | OFS Algorithm | | | Classification algorithm |
|---|---|---|---|---|
| | FwS | BwS | HcS | |
| GR | 94.43 | 96.11 | 97.29 | RF |
| IG | 95.23 | 93.24 | 96.65 | |
| ChS | 94.19 | 95.31 | 94.27 | |
| GR | 83.12 | 79.76 | 91.48 | DT |
| IG | 89.45 | 87.4 | 92.56 | |
| ChS | 98.19 | 92.56 | 93.73 | |
| GR | 68.75 | 96.15 | 85.34 | NN |
| IG | 89.48 | 66.09 | 90.65 | |
| ChS | 91.56 | 69.72 | 88.59 | |

**Table 5 Comparative results of best hybrid models.**

| Hybrid model | Sensitivity | Specificity | Precision | F1 score | AUC | MCC |
|---|---|---|---|---|---|---|
| GR – HcS – RF | 0.92 | 0.89 | 0.91 | 0.93 | 0.98 | 0.97 |
| ChS – FwS – DT | 0.99 | 0.98 | 0.99 | 0.99 | 0.99 | 0.98 |
| GR – BwS – NN | 0.93 | 0.94 | 0.98 | 0.88 | 0.98 | 0.91 |

highest accuracy of 98.19% with chi-squared and forward search techniques among all other hybrid techniques employed in this research. Lastly, the hybrid NN models offer an accuracy range between 62.71% to 96.15%.

When assessing the effectiveness of a hybrid model in a multiclass problem, relying solely on accuracy is insufficient (*Alpaydin, 2010*). It is crucial to consider other significant parameters such as recall, specificity, precision, negative predicted value of a particular class, AUROC, and F1-score of the predictive method (*Powers, 2020*). Table 5 provides a comprehensive comparison of these parameters for the best hybrid models where-in the decision tree model combined with the chi-squared and forward search techniques demonstrates superior results across all parameters, exhibiting an impressive F1-score and AUROC value of 0.99. Despite the hybrid random forest model outperforming the decision tree model in terms of accuracy, the latter showcases significantly better results across all other parameters. Notably, all hybrid random forest models exhibit suboptimal prediction of SARS-CoV-2 TCEs, while the hybrid decision tree approach excels.

Analyzing the reliability of the technique is crucial to determine susceptibility to over-fitting or under-fitting issues. Over-fitting occurs when the model performs well with the training data but fails to generalize to testing data while under-fitting results in poor performance on both the training and testing data. To ensure the reliability and consistency of the hybrid techniques used in this study, 5-fold cross-validation (5 FCV) was conducted for the top three hybrid techniques—RF, DT, and NN. The accuracies achieved by these top-performing hybrid models on different folds are presented in Table 6, and their accuracy plot is depicted in Fig. 3 which clearly indicates that the hybrid

| Table 6 5FCV results in terms of accuracy of three top hybrid models. | | | |
| --- | --- | --- | --- |
| Fold | ChS – FwS – DT | GR – HcS – RF | GR – BwS – NN |
| 1 | 98.19 | 97.21 | 93.88 |
| 2 | 97.89 | 98.14 | 95.65 |
| 3 | 98.97 | 95.45 | 97.01 |
| 4 | 98.23 | 96.82 | 95.45 |
| 5 | 97.11 | 98.16 | 94.29 |
| Mean accuracy | 98.078 | 97.156 | 95.256 |

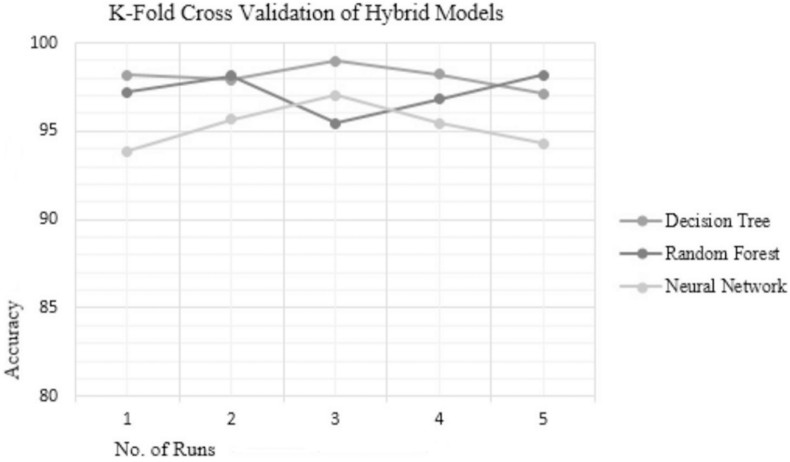

**Figure 3 KFCV results of hybrid models.**

DT model exhibits the most consistent accuracy results compared to the other hybrid techniques. The existing methods like the NetMHC server, as explained in the background section follow a probabilistic approach and estimate the peptide-binding capacity. While another method namely, the CTLpred server adopts a deterministic approach for prediction, it is constrained by the limited length of peptide sequences, restricted to 9-mers only. Hence, to address these challenges, a novel approach for TCE prediction following the deterministic approach is proposed herein, effectively resolving the issue associated with the NetMHC server as is evident through the results. In addition, the proposed hybrid model demonstrates the capability to predict epitopes of varying lengths (length > 9-mers), thereby overcoming the limitation associated with server CTLpred.

## Discussion

This study addresses a critical need in the field of vaccine development against SARS-CoV-2 by proposing novel hybrid ML techniques for predicting TCEs. The significance of our research lies in its potential to accelerate the identification of vaccine targets, thereby contributing to the global efforts to combat COVID-19. By leveraging computational methods to predict TCEs, our study offers a cost-effective and time-efficient approach compared to traditional laboratory-based methods. Furthermore, our proposed techniques

can predict peptides of variable lengths, including those longer than 9-mers, addressing a limitation of existing prediction techniques. Our findings not only advance the field of immunoinformatics but also hold promise for the development of PBVs against SARS-CoV-2. The deterministic nature of our predictive model distinguishes it from existing methods, which often rely on estimating binding potential rather than providing deterministic predictions. Moreover, the reliability and accuracy of our model, particularly when coupled with decision tree algorithms and feature selection techniques such as chi-squared and forward search, underscore its potential utility in vaccine design. In addition, the proposed hybrid ML models demonstrate robust performance across various evaluation metrics, underscoring their potential as valuable tools for accelerating the discovery and development of PBVs. Beyond COVID-19, the methodology presented in this study can be adapted and applied to other infectious diseases, providing researchers with a versatile approach for predicting TCEs and informing vaccine design efforts. The proposed work contributes to advancing the understanding of immune responses to viral pathogens and offers practical insights that could facilitate the development of next-generation vaccines with improved efficacy and specificity In nutshell, the study demonstrates the efficacy of hybrid ML approaches in predicting TCEs of SARS-CoV-2 and highlights their significance in expediting vaccine development. While further validation through experimental means *i.e.*, *in vivo* and *in vitro* is warranted, our research lays the foundation for future investigations into advanced ML classifiers and additional physicochemical properties to enhance vaccine design strategies.

## CONCLUSION

The frequent genetic changes and mutations that the SARS-CoV-2 virus underwent caused widespread concerns all over the world (*Harvey et al., 2021*). While several types of vaccines have been developed to address the disease, the PBVs have not received as much attention. However, these vaccines have several potential benefits, including increased safety, immunogenicity, and cost-effectiveness (*Yang et al., 2022*). Computational methods can be used to identify potential vaccine candidates more quickly and cost-effectively than traditional laboratory-based approaches (*Oluwagbemi et al., 2022*). In this study, we devised and assessed 27 hybrid machine learning techniques aimed at predicting T-cell epitopes of the SARS-CoV-2 virus. Following the extraction of features from the peptide sequences, we employed heuristic and greedy search techniques to pinpoint the optimal features for training the models. The evaluation of the models encompassed various performance metrics, including accuracy, sensitivity, specificity, and AUROC value. Our findings indicate that the decision tree, coupled with chi-squared and forward search, emerged as the most accurate and reliable predictive method. Our proposed model is unique in that it can deterministically predict TCEs, unlike other prediction techniques like NetMHC and CTLpred, which only estimate binding potential. Additionally, our model can predict peptides of various lengths, including those longer than 9-mers, which is a limitation of CTLpred. However, it is important to note that the predictions made by the model will still need to be validated through experimental means before being considered for use in a vaccine (*Humayun et al., 2022*). In conclusion, the proposed hybrid ML

techniques achieved outstanding results and outperformed state-of-the-art ML methods for TCE the prediction of TCEs of SARS-CoV-2 pathogen. Although PBVs have not yet been approved for human use, future research should focus on investigating other physicochemical properties and developing models using other advanced ML classifiers to improve the accuracy and other metrics. Overall, the use of computational methods to identify potential vaccine targets has the potential to have a significant impact on global health by saving lives, preventing future outbreaks, and reducing the potential for the virus to evade immunity through genetic mutations.

## Glossary of Terms

To facilitate a better understanding of the biological concepts discussed in this manuscript, we provide brief definitions or summaries of key terms relevant to immunology, virology, and bioinformatics. These definitions aim to clarify fundamental concepts and terminology used throughout the study, ensuring that readers, both experts and non-experts in the field, can engage with the scientific content effectively. The following definitions are provided for terms frequently referenced in the context of our investigation: amino acid, peptide, epitope, T-cells, vaccine, peptide-based vaccines, SARS-CoV-2 and COVID-19.

**Amino acid:** Building blocks of proteins; characterized by an amino group, carboxyl group, and a variable side chain.

**Peptide:** Short chains of amino acids linked by peptide bonds; serve various biological functions including signalling and enzyme activity.

**Epitope:** Specific region on an antigen recognized by the immune system; crucial for antibody binding and immune response initiation.

**T-cells:** A type of white blood cell critical for cell-mediated immunity; involved in recognizing and destroying infected or abnormal cells.

**Vaccine:** Biological preparation that stimulates the immune system to produce immunity to a specific disease, typically by introducing weakened or killed pathogens or their antigens.

**Peptide-based vaccines:** Immunizations composed of specific peptide sequences derived from antigens, capable of inducing an immune response against pathogens or diseases.

**SARS-CoV-2:** A novel coronavirus responsible for the COVID-19 pandemic, characterized by its ability to cause severe respiratory illness in humans.

**COVID-19:** A highly contagious respiratory illness caused by the SARS-CoV-2 virus, characterized by symptoms ranging from mild to severe, including fever, cough, and difficulty breathing, with potential complications such as pneumonia and acute respiratory distress syndrome (ARDS).

### Funding

This work was supported by the Deanship of Scientific Research at King Khalid University (KKU) through the Research Group Program Under the Grant Number: (R.G.P.2/572/44). The funders had no role in study design, data collection and analysis, decision to publish, or preparation of the manuscript.

### Grant Disclosures

The following grant information was disclosed by the authors:
Deanship of Scientific Research at King Khalid University (KKU) through the Research Group Program: R.G.P.2/572/44.

### Competing Interests

The authors declare that they have no competing interests.

### Author Contributions

- Syed Nisar Hussain Bukhari conceived and designed the experiments, performed the experiments, analyzed the data, performed the computation work, prepared figures and/or tables, authored or reviewed drafts of the article, and approved the final draft.
- E. Elshiekh conceived and designed the experiments, prepared figures and/or tables, authored or reviewed drafts of the article, and approved the final draft.
- Mohamed Abbas conceived and designed the experiments, prepared figures and/or tables, authored or reviewed drafts of the article, and approved the final draft.

### Data Availability

The source code and datasets are available at Zenodo: Syed Nisar, H. B. (2024). Physicochemical properties-based hybrid machine learning technique for the prediction of SARS-CoV-2 T-cell epitopes as vaccine targets. Zenodo. https://doi.org/10.5281/zenodo.10728289.

### Supplemental Information

Supplemental information for this article can be found online at http://dx.doi.org/10.7717/peerj-cs.1980#supplemental-information.

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
