# Peer review of "Physicochemical properties-based hybrid machine learning technique for the prediction of SARS-CoV-2 T-cell epitopes as vaccine targets"

_PeerJ Computer Science, doi:10.7717/peerj-cs.1980_

## Round 0.1 · original submission · Major Revisions

We have completed the 1st round review. Reviewers find merit in the manuscript and have suggested a major revision. You are required to address all the comments and suggestion of reviewers and submit a revision. Some of my comments are as follows:
1) The significance of the study needs to be discussed.
2) I would recommend replacing the term "hybrid machine learning" in the title with the extract name of the hybrid algorithm.
3) Cite a few relevant and recent papers in the Introduction section.

**Language Note:** The review process has identified that the English language must be improved. PeerJ can provide language editing services - please contact us at copyediting@peerj.com for pricing (be sure to provide your manuscript number and title). Alternatively, you should make your own arrangements to improve the language quality and provide details in your response letter. – PeerJ Staff

Reviewer 1 ·

Basic reporting

a) Could you elaborate on the significance of predicting TCEs (T-Cell Epitopes) of the SARS-CoV-2 virus and its relevance to the design of a PBV (Peptide-Based Vaccine)?

b) Extend introduction section by adding latest updates about the COVID-19.

Experimental design

a) Could you elaborate on the hybrid techniques mentioned and how they were implemented on the high-dimensional dataset?

b) Was there any specific emphasis on determining the most effective hybrid technique, and if so, why?

c) How did the researchers ensure a thorough and unbiased comparison among the various hybrid methodologies?

d) Were there any challenges or considerations specific to working with a high-dimensional dataset in this investigation?

e) Were there any notable trends or patterns observed during the comparative analysis of hybrid methodologies?

f) What are the three distinct Feature Weighting (FW) techniques, three Optimal Feature Selection (OFS) techniques, and three classification algorithms that are part of the methodology for predicting TCEs?

Validity of the findings

a) How do the findings from this investigation contribute to the understanding or advancement of knowledge in the field related to SARS-CoV-2 peptide sequences?

b) What is the role of feature selection (FS) in the machine learning (ML) process, and how does it contribute to the construction of a predictive model?

c) What criteria were considered during feature selection, and how do the chosen features contribute to enhancing the predictive accuracy of the model while addressing issues like computational cost, overfitting, and generalization?

Reviewer 2 ·

Basic reporting

The manuscript lacks proper English, and it is highly advised to complete the proofreading with professional services / fluent English support speakers.
The Literature Review should introduce the State of Art Section in an organized way.
The figures submitted by the authors are unclear and per the Journal Guidelines. The figures should be modified for clarity of vision and understanding into vector graphics.
Several terms in the Formal results do not include the definitions of the terms used in the equations and theorems.
It is highly recommended that these formulae be updated with proper terms and meanings.

Experimental design

The study amalgamates several ML techniques. However, the authors must present a brief introduction of each technique.
The authors imply the availability of 27 techniques due to Permutations of several ML algorithms. However no analytical treatment is presented by them towards adapting these techniques. There are several more eligible technique to apply for more better results, however no clear proof is presented for selecting them. They are expected to provide a proper understanding of this technique and its applicability.
Biological terms are generally used without proper definition/summary. The authors are advised to integrate the definition for a better reading experience of the manuscript.
In line 188, it's mentioned that the two CSV files were merged. However, there is no clear understanding of why the CSVs were merged. The authors must update the manuscript with a clear reason behind such initiatives in the article.
The authors have not mentioned proper reasoning to ensure the use of Feature Weighing with IG, GR and ChS. They must mention the criteria behind the choice.

Validity of the findings

The author calculated the AUROC and Mathews Correlation Coefficient. However, no clear indication is provided about the selection of these factors. The authors are advised to mention their criteria for selecting these values.

It is recommended that the authors include the definitions and proper mathematical explanations of the terms used in the study, such as Equations 1, 2, 3, 4, 5, 6, 7, 8, etc.

The experimental setup needs more elaboration of the terms used in the article for the readers to understand the paper more clearly.

The methodology must be explained in a more detailed way to help the reader understand the parallel recognition approach in the given study.

Various terms are not explained for the readers to make it understandable. It is recommended that the same is explained for all the terms used in the manuscript.
The novelty of research by the authors should be proved by a comparative study of all the State-of-the-art methods.

Additional comments

Several new citations and references from recent inventions should be included in the manuscript.
Kindly ensure the Table Citations and Figure Referencing appear in an ascending sequence in the manuscript.
All the figures under the results section must explain the findings and then compare them with the previous models used for similar kinds of study as per the literature review.
The paper contains potential, but with the aforesaid changes, it can be more beneficial and acceptable.
Authors must complete the reviews and modify the findings per the above sections.
They can resubmit the paper with the revisions. However, with the present format, the manuscript may not be suitable for the Journal's Scope and guidelines.

·

Basic reporting

Overall, this is an interesting study and results obtained are good. I would like the authors to address the following points in the revised manuscript:
When feature extraction was done, was class (epitope or non-epitope) added to CSV files?
What indicator has been used that suggest that the model is robust?
Author needs to clarify whether selected peptides used in the study are experimentally determined or predicted.
To which class do the peptide sequences belong? Do they belong to MHC I or MHC II or both? Kindly confirm the same and mention in the article.
What data partitioning rule was followed while model building?
Clarify that the dataset obtained from IEDB consists of linear or non-linear or both types of peptide sequences (ZIKV T-cell epitopes and non-epitopes)?
Language improvement is needed.

Experimental design

The authors should use the blind dataset to validate their model.
The authors can use multiple machine learning techniques to cross check the proposed model so it will
improve the result.

Validity of the findings

Highlight the applications and utility of the work.

---

## Round 0.2 · accepted · Accept

The authors have addressed all the comments and suggestions. I recommend that the manuscript be accepted for publication in its current form.

Reviewer 1 ·

Basic reporting

The authors have improved the manuscript as per my previous suggestions. No new suggestions. The manuscript can be considred for acceptance.

Experimental design

The authors have improved the manuscript as per my previous suggestions. No new suggestions. The manuscript can be considred for acceptance.

Validity of the findings

The authors have improved the manuscript as per my previous suggestions. No new suggestions. The manuscript can be considred for acceptance.

Additional comments

The authors have improved the manuscript as per my previous suggestions. No new suggestions. The manuscript can be considred for acceptance.

·

Basic reporting

I enjoyed reading this manuscript and believe that it is very promising.
Overall, this is an interesting study and results obtained are good.

Experimental design

Results obtained are good and promising and authors have incorporated all the changes in the revised manuscript.

Validity of the findings

This method is a reliable, rapid and useful prediction method.

Additional comments

The manuscript has been revised as per my comments and suggestions.